# Visualization of multidrug-resistant bacterial infection trends in the intensive care units

Woojae Jeon[1], Yu–Mi Lee[2,3], Dong Youn Kim[2], Ki–Ho Park[2], Mi Suk Lee[2,3]*

**1** College of Medicine, Graduate School, Kyung Hee University, Seoul, Republic of Korea, **2** Division of Infectious Diseases, Department of Internal Medicine, Kyung Hee University College of Medicine, Kyung Hee University Hospital, Seoul, Republic of Korea, **3** Korean National Healthcare–associated Infections Surveillance System Intensive Care Unit module, Seoul, Republic of Korea

* mslee7@gmail.com

## Abstract

### Background

The Korean National Healthcare–associated Infections Surveillance System (KONIS) monitors multidrug–resistant (MDR) bacterial infections in intensive care units (ICUs). However, simultaneously monitoring hundreds of ICUs remains challenging. Our study aimed to visualize the trends of MDR gram–negative bacterial infections in ICUs monitored by KONIS.

### Methods

We evaluated KONIS data from 137 ICUs (2006–2011) and from 368 ICUs (2017–2022). Pneumonia, urinary tract infection, and bloodstream infection caused by *Klebsiella pneumoniae*, *Pseudomonas aeruginosa*, and *Acinetobacter baumannii* were analyzed. Transformation was employed to convert the infection rate graphs of each ICU into arrows. The length and angle of the arrows reflect changes in carbapenem susceptibility and infection rate, respectively. ICUs are categorized into red (rapid shift from susceptible to resistant bacteria and increased infection rate), yellow (slow shift from susceptible to resistant bacteria and decreased infections rate), and green (shift from resistant to susceptible bacteria) groups. The proportional changes in each ICU category were compared during the first and last five years of the study periods.

### Results

For *K. pneumoniae*, the proportion of red category ICUs increased (0% to 17%, p-value 0.586), while the proportions of yellow (33.3% to 7%, p-value 0.288) and green category ICUs (66.6% to 36%, p-value 0.290) decreased. For *P. aeruginosa*, the proportions of red (12% to 27%, p-value 0.016) and green category ICUs (38% to 46%, p-value 0.358) increased, while the proportion of yellow category ICUs decreased (8% to 2%, p-value 0.043). For *A. baumannii*, the proportions of red

**Data availability statement:** Data cannot be shared publicly because of potentially sensitive information and the terms under which the data were collected indicated the data were to be used only for the purposes outlined in this study. Data may be available from KONIS ICU Steering Committee Office (contact via konis2@koshic.org) for researchers who meet the criteria for access to confidential data. (at the revision, synthetic data has been submitted)

**Funding:** The author(s) received no specific funding for this work.;

**Competing interests:** NO authors have competing interests

(19% to 14%, p-value 0.649) and yellow category ICUs (5% to 1%, p-value 0.187) decreased, while the proportion of green category ICUs increased (19% to 72%, p-value <0.001).

## Conclusions

Graph transformation allowed the observation of MDR Gram-negative bacterial infection trends in ICUs. Further studies should aim to confirm whether our arrow indicators are useful for infection control and in identifying factors for reducing infections.

## Introduction

Nosocomial and multi-drug resistant bacterial infections have increased significantly. As of 2022, *Acinetobacter baumannii* accounts for more than 50% of carbapenem-resistant gram-negative bacilli isolated in intensive care units in Korea, followed by *Klebsiella pneumoniae* and *Pseudomonas aeruginosa* accounting for more than 20% and 18%, respectively [1].

Since 1996, Republic of Korea has adopted nationwide medical surveillance following the standards set by the United States National Nosocomial Infections Surveillance System (NNIS). In 2006, Korean National Healthcare–associated Infections Surveillance System (KONIS) was officially launched. The KONIS is a validated and reliable national surveillance system that has been used for trend analysis and surveillance of nosocomial infections, intensive care unit (ICU) infections, and device–related infections [2–5]. The KONIS data also contain information on causative pathogens and their antibiotic susceptibility, and several studies using this information have been conducted [2,6,7]. It also provides information about both fungal and bacterial infections. Even during the coronavirus disease 2019 (COVID–19) pandemic, active surveillance was conducted using the KONIS. Research was performed to determine the impact of COVID–19 on healthcare–associated infections in the ICU [8]. However, most previous studies have focused on aggregated data or infection types, with limited investigation into infection and resistance trends at the individual ICU level.

The KONIS is used to monitor trends in specific site and device–related infections and resistant bacteria in the ICU. The burden of MDR Gram–negative bacterial infections in South Korea has increased significantly, as evidenced by the rise in imipenem resistance of *Pseudomonas aeruginosa* from 13.9% to 30.8% and *Acinetobacter baumannii* from 7% to 73.5% between 1997 and 2016 [9]. Given that more than 300 ICUs are monitored in KONIS, with varying surveillance durations extending up to 16 years, it remains difficult to effectively assess and compare ICU-specific infection trends. This complexity hinders real-time interpretation and the development of tailored infection control strategies at the ICU level.

Therefore, this study aimed to fill this gap by visualizing trends in infections and MDR gram-negative bacterial infections across individual ICUs using KONIS data. To our knowledge, no prior research has systematically evaluated or visualized antimicrobial resistance trends at the individual ICU level using KONIS data. By applying

a novel graph transformation method, we sought to condense complex surveillance results into interpretable patterns, enabling more intuitive comparisons and insights into ICU-specific resistance trends over time.

## Methods

### Setting and study design

This study was based on surveillance data from the Korean National Healthcare–associated Infections Surveillance System (KONIS), which monitors healthcare-associated infections (HAIs) in intensive care units (ICUs) nationwide. Participation in KONIS is voluntary and limited to medical institutions operating ICUs with ≥100 inpatient beds. As of 2022, 357 ICUs from 274 hospitals participated, representing approximately 85.9% of eligible institutions in Korea.

KONIS data are reported through the Korea Disease Control and Prevention Agency's (KDCA) Integrated Disease and Health Management System, a secure internet-based platform. Each institution designates authorized personnel who register, authenticate with digital certificates, and submit HAI data directly to the KONIS system. While basic statistical summaries are accessible to all registered users, raw data use for academic purposes requires approval by the KONIS Steering Committee. This study was approved for use of raw data, which were provided in de-identified Excel format.

We selected two time periods for analysis: 2006–2011 and 2017–2022. These were chosen based on changes in KONIS diagnostic criteria and expansion of participating institutions. In particular, diagnostic standards for urinary tract infections were revised in 2012 and 2016, and laboratory-based diagnostics for bloodstream infections and pneumonia were introduced in 2016. Additionally, the number of participating ICUs grew rapidly after 2016, enabling more stable data comparison across ICUs in the selected periods.

Among all HAIs monitored in KONIS, we focused on urinary tract infection, bloodstream infection, and pneumonia, which are commonly associated with invasive device use. The study analyzed *Klebsiella pneumoniae*, *Pseudomonas aeruginosa,* and *Acinetobacter baumannii*, which have shown rising incidence and resistance rates in Korean ICUs in recent years. In contrast, *Staphylococcus aureus* and *Enterococcus spp*., which have shown declining or stable trends, were excluded from this analysis.

Although the present year is 2025, we utilized KONIS data from 2022 because annual datasets are finalized and publicly reported 12–18 months after data collection ends. The study was planned in 2023 based on the most recently available and complete dataset at that time.

### Pathogen and disease

In this study, a separate analysis was conducted based on the pathogen identified (*Klebsiella pneumoniae, Pseudomonas aeruginosa,* and *Acinetobacter baumannii*). The bacterial infections were further stratified into carbapenem–susceptible (CS) and carbapenem–resistant (CR) types based on carbapenem susceptibility. The susceptibility of bacteria to carbapenems was determined according to the guidelines of the Clinical and Laboratory Standards Institute (CLSI) [10]. During the study period, the susceptibility breakpoints for imipenem, meropenem, and doripenem in *K. pneumoniae* were revised from ≤4 to ≤1 mg/L in 2010. For *P. aeruginosa*, the susceptibility breakpoints were revised from ≤4 to ≤2 mg/L in 2012. Similarly, the susceptibility breakpoints for *A. baumannii* were revised from ≤4 to ≤2 mg/L in 2014 [10,11].

The diseases analyzed were pneumonia, urinary tract infection (UTI), and bloodstream infection (BSI). Because the risk of healthcare-associated infections among patients hospitalized for less than 3 days was reported to be below 1% in previous KONIS validation studies, these patients were excluded from the analysis [12]. Pneumonia was characterized by new, persistent, or worsening findings (infiltration, consolidation, and cavitation) on two or more consecutive chest radiographs. Additionally, a diagnosis was established if fever (>38°C), leukopenia (≤4,000 white blood cell/mm$^3$), leukocytosis (≥12,000 white blood cell/mm$^3$), or altered mental status was noted without other identified causes in older age (≥70 years) and two of the following criteria were present: new purulent sputum, changes in sputum pattern, increased

respiratory secretions, or increased need for suction; new or worsening cough, dyspnea, or tachypnea; rales or crackles; or worsening gas exchange, increased oxygen demand, or increased ventilation requirements.

Urinary tract infection was diagnosed if one of the following criteria was present: fever (>38°C), suprapubic tenderness, costovertebral angle pain, frequency, urgency, dysuria, and ≤2 types of bacteria identified in urine culture and at least 1 type of bacteria isolated at $10^5$ colony forming units/ml or more.

Blood stream infection was defined as the presence of one or more pathogenic bacteria on blood culture or non–culture–based testing (polymerase chain reaction, matrix–assisted laser desorption/ionization time–of–flight mass spectrometry, and rapid antigen test on blood samples); the isolated bacteria were not related to infection in other areas. If skin normal flora was isolated from two or more independently collected blood samples, BSI was diagnosed if the bacteria detected from the blood samples were unrelated to infection in other areas, and if fever (>38°C), chills, or hypotension occurred.

## Intensive care units

Among the ICUs that participated in KONIS from 2006 to 2022, we selected those that participated in the first five years (2006–2011) and those that participated in the last five years (2017–2022). KONIS has collected and analyzed data on a quarterly basis according to its own guidelines [2,3]. The first and last periods may provide more consistent and comparable data sets because, in the middle periods, there were some changes in KONIS guidelines aimed at improving data collection. From this pool, we further refined our selection to include ICUs where infections (pneumonia, UTI, and BSI) due to *K. pneumoniae*, *P. aeruginosa*, and *A. baumannii* occurred in at least one quarter. In the selected ICUs, the bacterial infections were classified as CS and CR types according to carbapenem susceptibility. ICUs where both CS and CR bacterial infections had occurred for more than two quarters were finally selected to calculate the change of bacterial susceptibility.

## Process of graph transforming

From a macroscopic perspective, it was hypothesized that a physical force exists which transforms the susceptible bacterial population into a resistant bacterial population. This force was believed to reflect various factors, including antimicrobial stewardship and infection control. Since vectors are commonly used to represent force in physics, this concept was applied in the study. In physics, when a force acts on an object, it is considered to act on the center of mass of that object. Similarly, the force was considered to act on the distribution centers of the susceptible and resistant bacterial populations. To create the vectors, we performed the following process. First, the overall rate of infections (pneumonia, UTI, and BSI) that occurred during the observation period in each ICU was plotted according to the pathogen involved (AB, KP, and PA). These rates were then delineated according to the carbapenem susceptibility status (Fig 1. A, D, and G). The overall infection rate was calculated as follows: number of infections (pneumonia, UTI, or BSI)/ total patient–days × 1000. Using the overall infection rate graph, the average CS and CR bacterial infection rates during the observation period were plotted together with the error range for each ICU (Fig 1. B, E, and H). An arrow was drawn with the average of the CS bacterial infection rate as the starting point and that of the CR bacterial infection rate as the endpoint. Then, the length and angle of the arrow were measured. In this process, the overall infection rate was scaled up 20 times for the optimal readability (S1 File). Through these processes, the overall infection rate versus time graph for each ICU was transformed into arrows indicating length and angle. Additionally, by displaying only the endpoint of the arrow in polar coordinates, the overall infection rate versus time graph for each ICU was condensed into a single dot (Fig 1. C, F, and I). The several equations were used for vector calculation (S2 File).

Fig 1 explains the method of representing the overall infection rate graph over time at the individual ICU level by sequentially transforming it into a single vector. Through three examples, it illustrates the three vector categories, which will be discussed later.

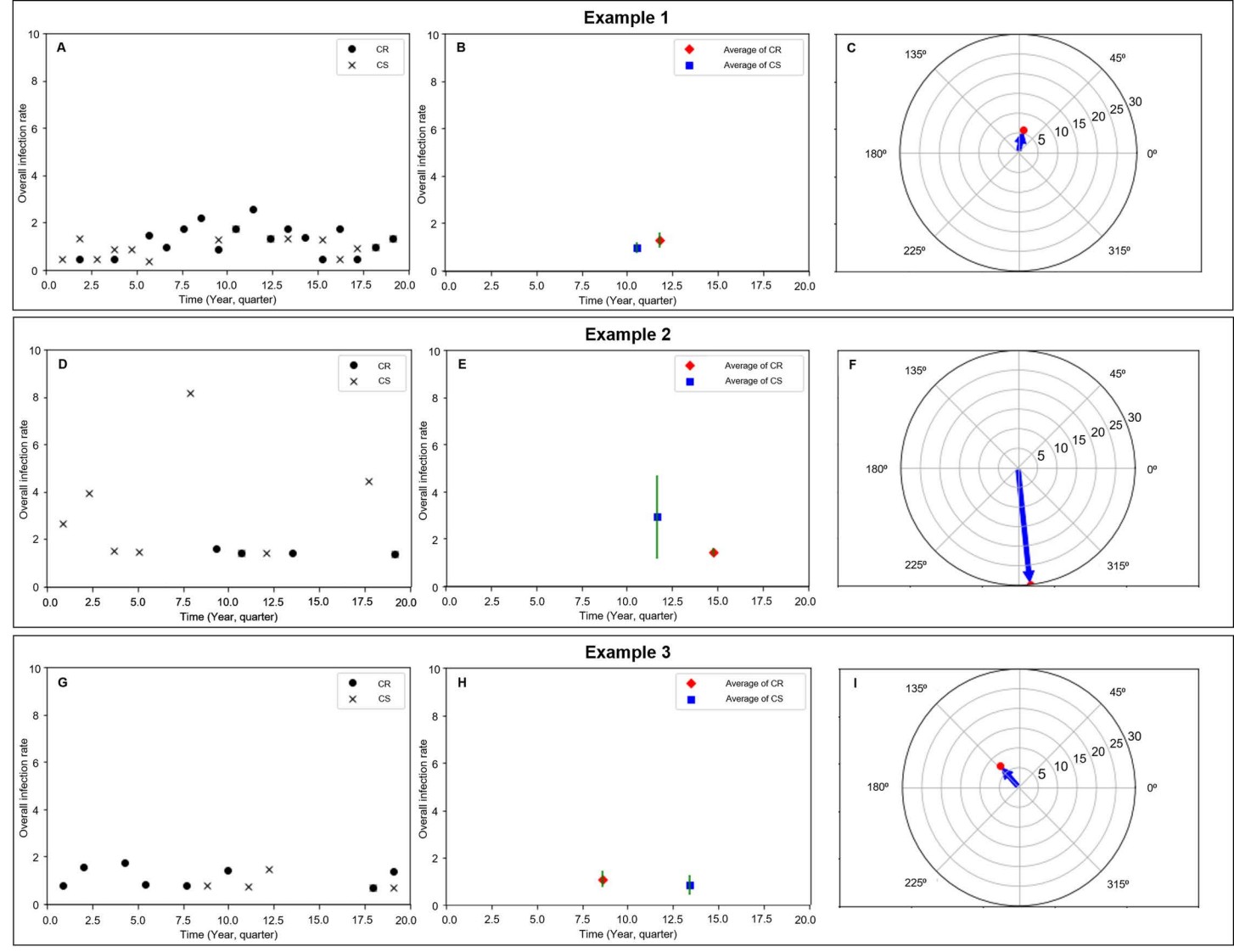

**Fig 1. Process of graph transform. A, D, G:** The overall infection rate (pneumonia, UTI, and BSI) at the individual ICU. **B, E, H:** The distribution center of carbapenem-susceptible (CS) and carbapenem-resistant (CR) bacterial infection rates with an error range at the individual ICU. **C, F, I:** Arrows showing the transition from CS to CR infection rate distribution centers in individual ICUs.

## Angle and length of arrows

A left-sided arrow indicates a change from resistance to susceptibility, whereas a right-sided arrow indicates a change from susceptibility to resistance. The length of the arrow correlates with the speed of transition from antibiotic-susceptible bacteria to antibiotic-resistant bacteria. A shorter length indicates a rapid transition, while a longer length indicates a slow transition (Fig 2A). An upward arrow indicates an increase in the overall infection rate, while a downward arrow indicates a decrease. The closer the angle of the arrow is to 90°, the greater the increase in overall infection rates. Conversely, the closer the angle of the arrow is to 270°, the greater the decrease in overall infection rates. For example, an arrow with an angle of 60° indicates a greater increase in the overall infection rate than an arrow with an angle of 30° (Fig 2B).

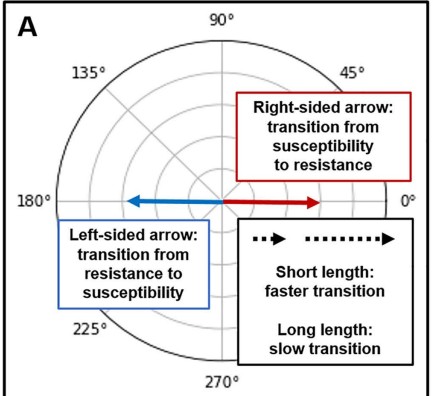 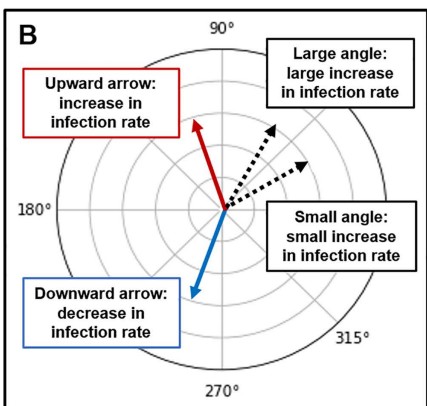 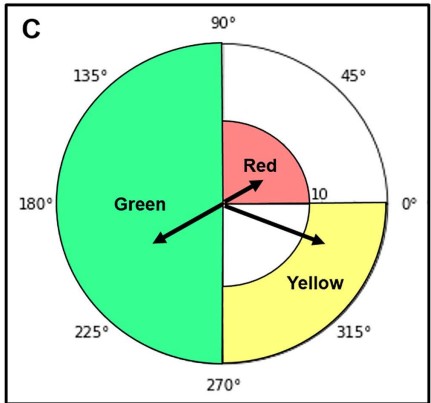

**Fig 2. Angle and length of arrow and ICU category. A:** A left-sided arrow indicates a shift from resistance to susceptibility, whereas a right-sided arrow indicates a change from susceptibility to resistance. The length of the arrow inversely correlates with the transition speed from antibiotic-susceptible bacteria to antibiotic-resistant bacteria. **B:** An upward arrow indicates an increase in the overall infection rate, while a downward arrow indicates a decrease. **C:** ICUs are categorized into red (rapid shift from susceptible to resistant bacteria and increased infection rate), yellow (slow shift from susceptible to resistant bacteria and decreased infections rate), and green (shift from resistant to susceptible bacteria) groups.

### Intensive care unit categorization based on arrow indicators

Based on the angles and lengths of the arrows for each ICU, we have categorized ICUs into red, yellow, and green groups (Fig 2C). The red category (aggressive ICU group) was defined as ICUs where the bacteria underwent a rapid transition from susceptible to resistant strains accompanied by an increase in the infection rate. In the graph, the arrows were notably short (<10), and the angle fell between 0° and 90°. The yellow category (indolent ICU group) was defined as ICUs where the bacteria underwent a slow transition from susceptible to resistant strains accompanied by a simultaneous decrease in infection rates. In the graph, the arrows were notably long (≥10), and the angle fell between 270° and 360°. The cutoff length of transition speed used to define the red and yellow categories was set at 10, which is approximately the average length of all arrows. Lastly, the green category (reverse ICU group) was defined as ICUs where bacteria exhibited a reversed trend, from resistant to susceptible strains. In the graph, the angle of the arrow fell between 90° and 270°, regardless of length. Angle and length of Fig 2 explains the meaning of the vector's length and angle. It also aims to categorize the ICUs into three groups based on the vector length and angle. The left arrow represents the change from resistance to susceptibility, while the right arrow represents the change from susceptibility to resistance.

### Ethics approval and consent to participate

All participated intensive care unit's data were fully anonymized before we accessed them. Since this epidemiological study using retrospective, deidentified KONIS surveillance data set, the Institutional Review Board of Kyung Hee University Hospital conducted exempt review and waived the requirement for informed consent (Seoul, Republic of Korea; IRB No. KHUH 2023–12–020).

### Statistical analysis

Python and Microsoft Excel were used for the overall data extraction process, classification, and graph drawing (S3 File and S4 File). In Tables 1 and 2, SPSS (Student's t-test, Mann-Whitney test, chi-square test, Fisher's exact test) was used. For comparing continuous data, Student's t-test was applied when the normality test conditions were met, and Mann-Whitney test was used when the normality test conditions were not satisfied. For comparing categorical data, chi-square test was used, and Fisher's exact test was applied when more than 20% of the cells had expected frequencies less than 5

**Table 1. Characteristics of ICUs included in the study.**

| Variable | A. baumannii | | | K. pneumoniae | | | P. aeruginosa | | |
|---|---|---|---|---|---|---|---|---|---|
| | First 5 years (2006–2011) | Last 5 year (2017–2022) | P value | First 5 years (2006–2011) | Last 5 year (2017–2022) | P value | First 5 years (2006–2011) | Last 5 year (2017–2022) | P value |
| Total ICU numbers | 43 | 102 | | 3 | 81 | | 50 | 92 | |
| Type of hospitals | | | | | | | | | |
| Public | 7 (16.3) | 19 (18.6) | 0.736 | 0 (0.0) | 17 (21.0) | 0.99 | 9 (18.0) | 16 (17.4) | 0.928 |
| Private | 36 (83.7) | 83 (81.4) | | 3 (100.0) | 64 (79.0) | | 41 (82.0) | 76 (82.6) | |
| Teaching hospital | 43 (100.0) | 85 (83.3) | 0.004 | 3 (100.0) | 71 (87.7) | 0.99 | 50 (100.0) | 80 (87.0) | 0.008 |
| Size of hospitals | | | | | | | | | |
| 100–299 beds | 0 (0.0) | 5 (4.9) | 0.001 | 0 (0.0) | 7 (8.6) | 0.662 | 0 (0.0) | 9 (9.8) | 0.001 |
| 300–499 beds | 0 (0.0) | 17 (16.7) | | 0 (0.0) | 16 (19.8) | | 2 (4.0) | 15 (16.3) | |
| 500–699 beds | 10 (23.3) | 25 (24.5) | | 1 (33.3) | 18 (22.2) | | 13 (26.0) | 20 (21.7) | |
| 700–899 beds | 17 (39.5) | 35 (34.3) | | 1 (33.3) | 19 (23.5) | | 15 (30.0) | 29 (31.5) | |
| ≥ 900 beds | 16 (37.2) | 20 (19.6) | | 1 (33.3) | 21 (25.9) | | 20 (40.0) | 19 (20.7) | |
| Infection control staff | | | | | | | | | |
| Infection specialist per 1000 beds | 1.65 [1.21–2.00] | 2.54 [1.51–3.14] | 0.001 | 1.65 [1.58–2.06] | 2.56 [1.64–3.15] | 0.282 | 1.62 [1.21–1.87] | 2.24 [1.01–3.12] | 0.002 |
| Infection control nurse per 1000 beds | 1.81 [1.34–2.15] | 7.25 [6.92–7.93] | <0.001 | 1.64 [1.51–2.07] | 7.27 [6.94–7.95] | <0.001 | 2.00 [1.41–2.71] | 7.28 [6.92–8.11] | <0.001 |
| Type of ICU | | | | | | | | | |
| Medical | 29 (67.4) | 68 (66.7) | 0.928 | 1 (33.3) | 55 (67.9) | 0.257 | 29 (58.0) | 61 (66.3) | 0.327 |
| Surgical | 14 (32.6) | 34 (33.3) | | 2 (66.7) | 26 (32.1) | | 21 (42.0) | 31 (33.7) | |
| Number of ICU beds | 18.00 [14.08–23.71] | 18.00 [14.98–23.16] | 0.921 | 17.00 [16.50–20.50] | 17.50 [14.00–22.70] | 0.716 | 20.00 [14.06–24.13] | 18.20 [14.85–24.15] | 0.996 |
| ICU nurse per ICU beds | 1.27 [1.13–1.56] | 1.82 [1.48–2.07] | <0.001 | 1.35 [1.27–1.36] | 1.50 [0.94–1.84] | 0.071 | 1.29 [1.16–1.66] | 1.84 [1.42–2.02] | 0.002 |
| Total patient–days in ICU, | 1710.00 [1361.35–2164.33] | 1366.28 [1078.8–1743.75] | <0.001 | 1756.75 [1696.88–2077.06] | 1330.50 [1091.83–1731.90] | 0.071 | 1804.36 [1332.98–2214.20] | 1366.28 [1090.54–1847.91] | 0.001 |

Data are presented as no. (%) or median [interquartile range] unless otherwise indicated.

Abbreviations: BSI = bloodstream infection, ICU = intensive care unit, KONIS = Korean National Healthcare–associated Infections Surveillance System, UTI = urinary tract infection.

# Results

## Screening, enrollment, and inclusion

A total of 137 ICUs participated in the first 5 years (2006–2011) and 368 ICUs participated in the last 5 years (2017–2022). During the first 5 years, KP infection was reported in 122 ICUs, PA infection in 124 ICUs, and AB infection in 123 ICUs. In the last 5 years, KP infection was reported in 316 ICUs, PA infection in 316 ICUs, and AB infection was reported in 309 ICUs.

Intensive care units (ICUs) where CS and CR bacterial infections occurred for more than two quarters were finally selected. A total of 3 ICUs with KP infection, 43 ICUs with AB infection, and 50 ICUs with PA infection in the first 5 years, and 81 ICUs with KP infection, 92 ICUs with PA infection, and 102 ICUs with AB infection in the last 5 years were included in the final analysis (Fig 3). The characteristics of ICUs included in this study were shown in Tables 1 and 2.

**Table 2. The rates of urinary tract infection, bloodstream infection, and pneumonia in ICUs.**

| Variable | A. baumannii | | | K. pneumoniae | | | P. aeruginosa | | |
|---|---|---|---|---|---|---|---|---|---|
| | First 5 years (2006–2011) | Last 5 year (2017–2022) | P value | First 5 years (2006–2011) | Last 5 year (2017–2022) | P value | First 5 years (2006–2011) | Last 5 year (2017–2022) | P value |
| **Urinary tract infection** | | | | | | | | | |
| UTI rate | 4.12 [2.68–4.82] | 1.23 [0.73–1.97] | <0.001 | 4.15 [3.97–4.49] | 1.61 [1.01–2.20] | 0.001 | 4.57 [3.62–5.50] | 1.50 [0.93–2.26] | <0.001 |
| Catheter–associated UTI rate | 4.66 [3.13–5.98] | 1.22 [0.72–1.90] | <0.001 | 4.93 [4.54–5.16] | 1.59 [0.98–2.18] | <0.001 | 5.17 [4.39–6.46] | 1.48 [0.92–2.23] | <0.001 |
| Urinary catheter utilization ratio | 0.88 [0.79–0.94] | 0.89 [0.85–0.92] | 0.178 | 0.89 [0.85–0.90] | 0.88 [0.83–0.91] | 0.964 | 0.88 [0.79–0.93] | 0.87 [0.82–0.92] | 0.640 |
| Number of urinary catheter–days | 1164.76 [868.86–1865.40] | 1181.80 [955.47–1492.17] | 0.002 | 1610.75 [1543.58–1774.33] | 1161.65 [930.15–1431.40] | 0.036 | 1592.78 [1124.18–1922.85] | 1186.93 [951.1–1565.70] | 0.001 |
| **Bloodstream infection** | | | | | | | | | |
| BSI rate | 1.96 [1.42–2.59] | 1.76 [1.19–2.89] | 0.659 | 2.20 [2.08–2.69] | 1.86 [1.13–3.29] | 0.566 | 1.84 [1.37–2.46] | 1.97 [1.08–3.10] | 0.799 |
| Central line–associated BSI rate | 3.40 [2.21–4.30] | 1.48 [1.01–2.59] | <0.001 | 3.76 [3.42–3.90] | 1.57 [0.91–2.75] | 0.024 | 2.91 [2.04–4.39] | 1.59 [0.89–2.65] | 0.000 |
| Central line utilization ratio | 0.51 [0.43–0.66] | 0.57 [0.48–0.68] | 0.307 | 0.58 [0.54–0.59] | 0.58 [0.45–0.69] | 0.910 | 0.52 [0.42–0.64] | 0.58 [0.46–0.69] | 0.248 |
| Number of central line–days | 895.00 [738.31–1192.40] | 784.10 [581.12–1013.56] | 0.027 | 1037.75 [925.88–1235.91] | 770.20 [538.13–997.28] | 0.149 | 869.46 [696.94–1236.95] | 773.48 [599.76–1029.41] | 0.071 |
| **Pneumonia** | | | | | | | | | |
| Pneumonia rate | 1.39 [0.87–1.88] | 0.70 [0.34–1.15] | <0.001 | 3.26 [2.06–3.40] | 0.82 [0.33–1.25] | 0.051 | 1.41 [0.80–2.27] | 0.83 [0.34–1.41] | <0.001 |
| Ventilator–associated pneumonia rate | 2.20 [1.39–4.05] | 0.34 [0.12–0.61] | <0.001 | 5.81 [3.36–6.23] | 0.32 [0.10–0.68] | 0.004 | 1.91 [1.24–4.10] | 0.40 [0.12–0.70] | <0.001 |
| Ventilator utilization ratio | 0.36 [0.28–0.48] | 0.37 [0.30–0.50] | 0.873 | 0.40 [0.32–0.47] | 0.43 [0.28–0.56] | 0.838 | 0.36 [0.27–0.49] | 0.40 [0.30–0.51] | 0.538 |
| Number of ventilator–days | 463.50 [364.17–929.50] | 546.88 [346.52–831.04] | 0.019 | 702.38 [552.89–1000.11] | 578.05 [327.58–817.45] | 0.445 | 622.69 [444.18–955.20] | 546.88 [341.99–840.33] | 0.064 |

Data are presented as no. (%) or median [interquartile range] unless otherwise indicated.

Abbreviations: BSI = bloodstream infection, ICU = intensive care unit, UTI = urinary tract infection.

infection rate = number of infections/ total patient–days × 1000

urinary catheter utilization ratio = number of urinary catheter–days/ number of patient–days

ventilator utilization ratio = number of ventilator–days/ number of patient–days

central line utilization ratio = number of central line–days/ number of patient–days

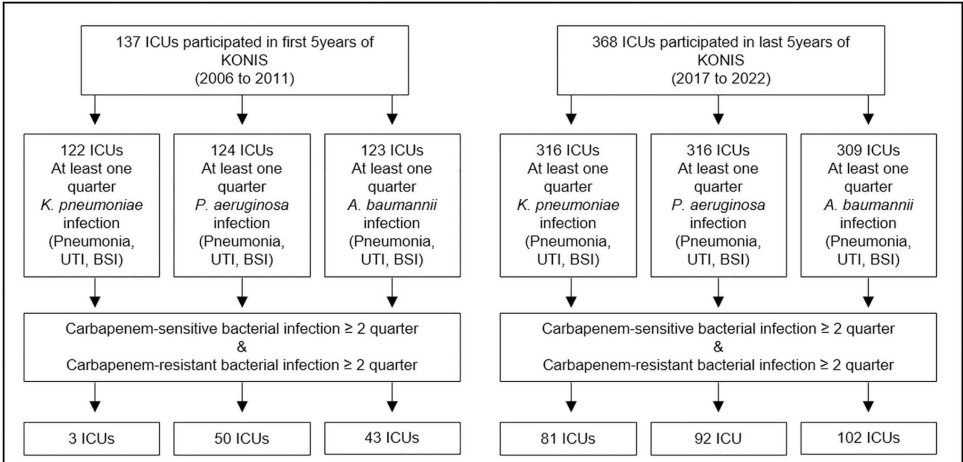

**Fig 3. Flowchart of the enrollment in this study.** BSI = bloodstream infection, ICU = intensive care unit, KONIS = Korean National Healthcare–associated Infections Surveillance System, UTI = urinary tract infection.

## The visualization of infection trends in Intensive care units

The graph below shows a collection of the arrow endpoints corresponding to the ICUs associated with each pathogen and observation period (Fig 4). Each ICU is represented as a single point, with all monitored ICUs simultaneously displayed on one graph. Vectors are drawn from the origin to each point, and each vector reflects the changes in the distribution of susceptible and resistant bacterial groups, as well as the infection rate, at each individual ICU.

## Arrow category trend in the first and last 5 years of KONIS

Based on the ICU distribution graph (Fig 4), we divided the ICUs into red, yellow, and green categories according to the length and angle of the arrow. For KP infection, the proportion of ICUs in the red category increased in the last 5 years compared to the first 5 years (p-value 0.586), while the proportions in the yellow and green categories decreased (p-value 0.288, p-value 0.290) (Fig 5C and D). For PA infection, both the red and green category ICUs increased in proportion (p-value 0.016, p-value 0.358), while the yellow category decreased during the last 5 years compared to the first 5 years (p-value 0.043) (Fig 5E and F). For AB infection, the proportion of ICUs in the green category increased (p-value <0.001)., whereas the red and yellow categories decreased in the last 5 years compared with the first 5 years (p-value 0.649, p-value 0.187) (Fig 5A and B).

The aim of Fig 5 was to present the proportion of ICUs in the red category, which require vigilant monitoring, and the proportion of ICUs in the green category, where infection control is being effectively managed, in a manner that allows for easy comprehension.

## Discussion

### Key findings and interpretation and unexpected findings

Compared with that in the first 5 years of the KONIS, the number of participating ICUs increased in the last five years, and the number of ICUs in which infections (pneumonia, UTI, and BSI) were caused by KP, PA, and AB also increased. According to the results obtained through a series of graph transformation processes, the trend of the resistant bacteria seemed to exhibit slight variations depending on the pathogen.

In the case of KP, the proportion of aggressive ICUs, characterized by a rapid transition from CSKP to CRKP with a simultaneous increase in infection rates, increased. Conversely, the proportion of indolent ICUs, marked by a slow

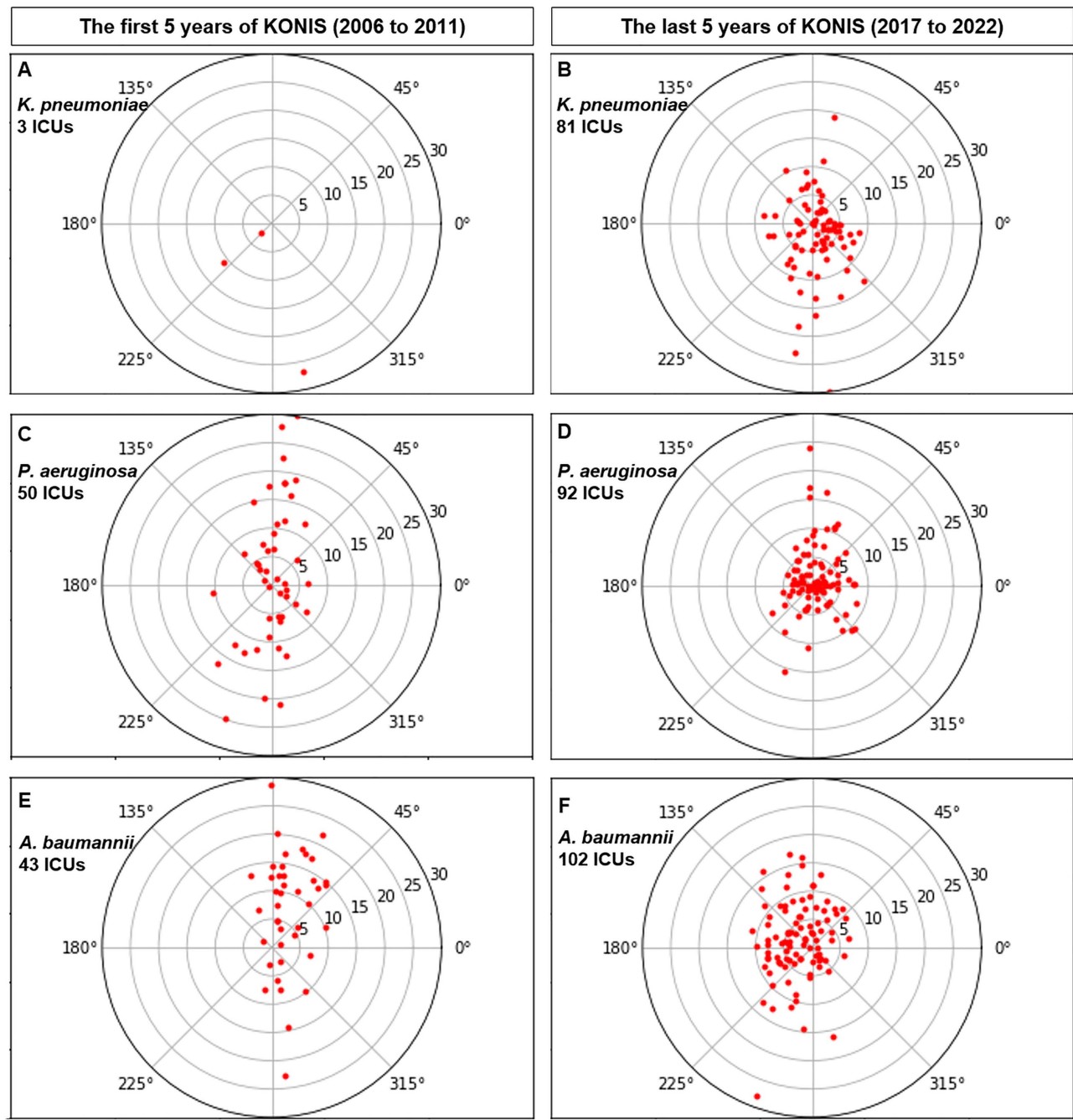

**The first 5 years of KONIS (2006 to 2011)**

**A** *K. pneumoniae* 3 ICUs

**C** *P. aeruginosa* 50 ICUs

**E** *A. baumannii* 43 ICUs

**The last 5 years of KONIS (2017 to 2022)**

**B** *K. pneumoniae* 81 ICUs

**D** *P. aeruginosa* 92 ICUs

**F** *A. baumannii* 102 ICUs

**Fig 4. Distribution of ICUs.** Arrow endpoints for individual ICUs are plotted in polar coordinates, with all starting points shifted to the origin. **A:** *K. pneumoniae* infection during the first 5 years. **B:** *K. pneumoniae* infection during the last 5 years: **C:** *P. aerugionosa* infection during the first 5 years. **D:** *P. aerugionosa* infection during the last 5 years. **E:** *A. baumannii* infection during the first 5 years **F:** *A. baumannii* infection during the last 5 years.

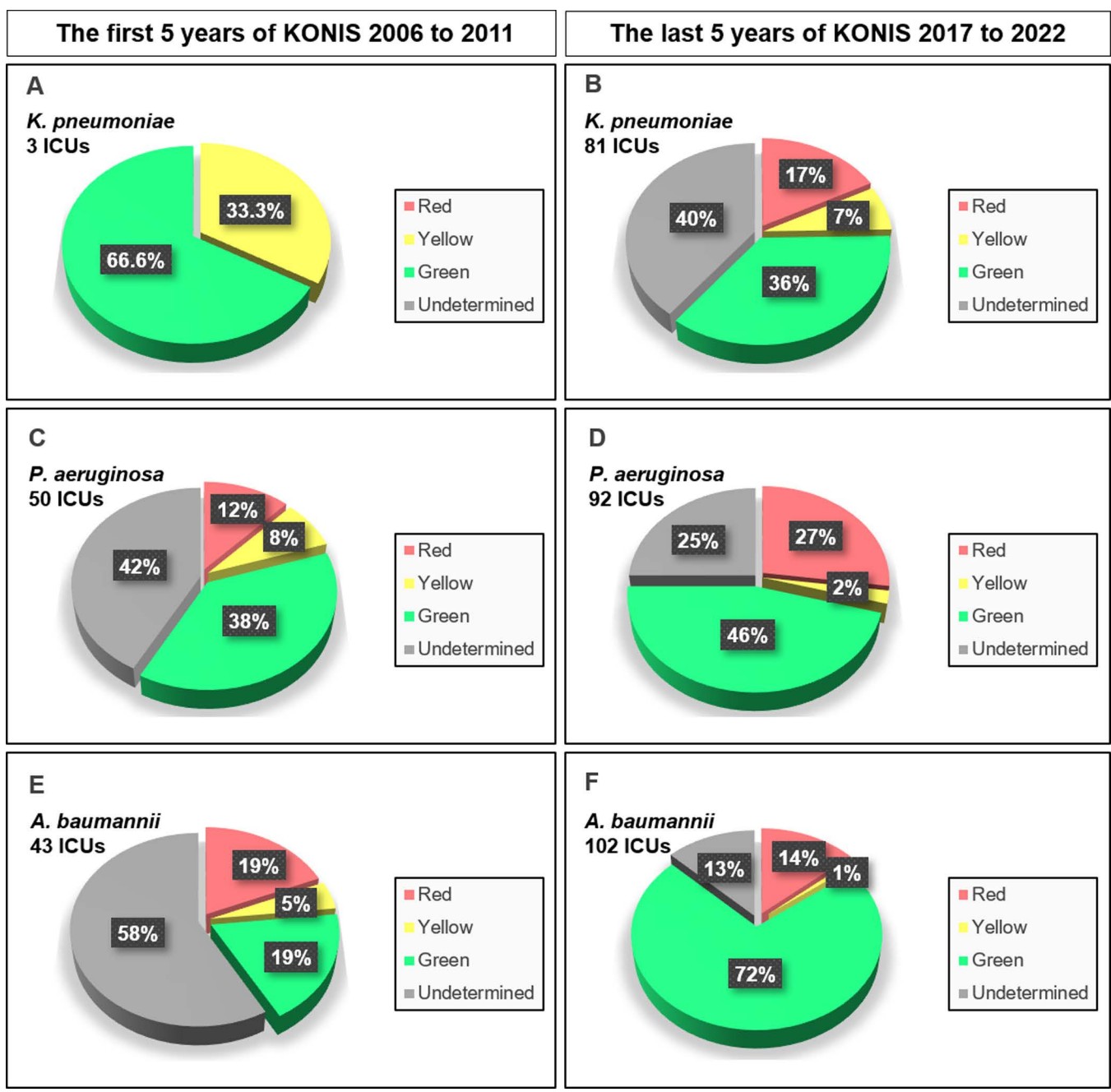

**Fig 5. Arrow category trend during the first and last 5 years of the KONIS. A:** *K. pneumoniae* infection during the first 5 years. **B:** *K. pneumoniae* infection during the last 5 years: **C:** *P. aeruginosa* infection during the first 5 years. **D:** *P. aerugionosa* infection during the last 5 years. **E:** *A. baumannii* infection during the first 5 years. **F:** *A. baumannii* infection during the last 5 years.

transition and simultaneous decrease in infection rates, and that of ICUs with a reversal of the infection trend from CRKP to CSKP decreased. Hence, these findings should be interpreted with caution due to the lack of data from the first 5 years; however, in terms of infection control, these results suggest a negative trend. A follow–up study is warranted to determine the ICU characteristics contributing to these outcomes.

In the case of PA, the proportion of aggressive ICUs, characterized by a rapid transition from CSPA to CRPA with a simultaneous increase in the infection rate, increased. Concurrently, the proportion of ICUs showing a reversal in the infection trend from CRPA to CSPA increased. This polarization phenomenon is an unexpected outcome. Therefore, a comparative analysis of the aggressive ICU characteristics and reverse ICU characteristics is needed to identify the key factors influencing this pattern. In the analysis of *P. aeruginosa*, the proportion of newly participating ICUs in the green and red categories in the last 5 years of KONIS, which initially did not participate in KONIS but joined in the later stages, was 78.5% and 76%, respectively. Most of the newly participating ICUs in the later stages of KONIS were small and medium-sized hospitals. It is speculated that the polarization observed may have been due to the lack of standardized infection control protocols, with significant differences between hospitals.

In the case of AB, the proportion of aggressive ICUs with a rapid transition from CSAB to CRAB and a simultaneous increase in infection rate decreased. Similarly, the proportion of indolent ICUs with a slow transition and simultaneously decreasing infection rate decreased. Conversely, the proportion of ICUs where the infection trend reversed from CRAB to CSAB increased. From an infection control perspective, these findings suggest a positive trend, and further studies are needed to determine the ICU characteristics contributing to these outcomes.

In the latter 5 years of KONIS, the proportion of medical ICUs increased compared to surgical ICUs. This may be due to the expansion of participation by small- and medium-sized hospitals, which typically have more medical than surgical ICUs. Additionally, increased national attention to infection control and antimicrobial stewardship, especially in high-risk medical ICUs with prolonged stays and frequent antibiotic use, may have encouraged greater participation from these units. These factors likely contributed to the observed trend shifts between the two periods.

## Significance and implications

Through the graph transformation processes presented in this study, trends in infections and resistant bacteria within individual ICUs over several years were condensed into a single arrow or dot. By plotting hundreds of ICUs as individual points on a single graph, we were able to intuitively visualize the transition tendencies from antibiotic-susceptible to antibiotic-resistant bacteria and changes in infection rates. Based on this visualization, ICUs were categorized into red, yellow, and green groups. We also compared the first and last five years of KONIS data to observe changes in group proportions over time.

This approach addresses key limitations of traditional methods, which rely on overall summary statistics or ICU specific trend graphs over many years, often impractical for large-scale surveillance. While systems such as GLASS (Global Antimicrobial Resistance Surveillance System) and NHSN (National Healthcare Safety Network) provide valuable aggregated data, they do not offer a granular view of resistance dynamics at the individual ICU level [13,14]. The new method allows a more intuitive, comprehensive, and scalable view of surveillance data, accommodating the growing number of participating ICUs and enabling clearer insights into unit-specific trends.

If sufficient data were available, it might be possible to approach the issue in the same manner by varying the combinations of pathogens and antibiotics/antivirals/antifungals. Also this could be done on a larger scale, such as at the national level, or on a smaller scale, such as at the hospital or even the ward level. However, selecting the appropriate reference values (e.g., cut-off length, scale-up factor) would likely be a crucial aspect of this approach.

## Limitations

Most importantly, the series of graph transformation processes introduced in this study are novel methods that have not been explored before and necessitate validation through comparison with conventional methods. Additional research

is required to determine the most appropriate scaleup factors for the overall infection rate when constructing the arrow. Although the quadrant to which individual ICUs belong remains consistent regardless of the scaleup factor, the polar coordinate system may exhibit congestion based on the chosen scaleup factor. Therefore, for convenient analysis, it is important to select an appropriate scaleup factor and maintain uniformity across comparison targets. Similarly, the selection of a suitable cutoff length for the arrow, delineating the red and yellow categories, holds significance, and standardization across comparison targets is advisable. The selection of an appropriate scaleup factor and cut–off length should align with the circumstances of the country or region where the analysis will be conducted, and this standardization should be consistent across comparison targets.

## Conclusions

Through the series of graph transformation processes presented in this study, it was possible to easily observe the trends of infections and resistant bacteria in hundreds of ICUs subjected to surveillance. Although the method employed in this study may seem unfamiliar due to its novelty, potential validation through future comparisons with conventional methods could position it as a new infection control indicator. However, to fully evaluate its usefulness, further validation studies and the judgment of readers will be necessary.

## Supporting information

**S1 File. Data distribution tendency according to scale-up factor.**
(DOCX)

**S2 File. Vector calculation formulas.**
(DOCX)

**S3 File. Python script for data analysis.**
(DOCX)

**S4 File. Synthetic dataset.**
(ZIP)

## Acknowledgments

We thank Korea Disease Control and Prevention Agency, KONIS ICU module, all participating ICUs and staff for their support.

## Author contributions

**Conceptualization:** Woojae Jeon, Dong Youn Kim, Yu-Mi Lee, Ki-Ho Park, Mi Suk Lee.

**Data curation:** Woojae Jeon, Dong Youn Kim, Yu-Mi Lee, Ki-Ho Park, Mi Suk Lee.

**Formal analysis:** Woojae Jeon, Dong Youn Kim, Yu-Mi Lee, Ki-Ho Park, Mi Suk Lee.

**Software:** Woojae Jeon, Dong Youn Kim, Yu-Mi Lee, Ki-Ho Park, Mi Suk Lee.

**Visualization:** Woojae Jeon, Dong Youn Kim, Yu-Mi Lee, Ki-Ho Park, Mi Suk Lee.

**Writing – original draft:** Woojae Jeon, Dong Youn Kim, Yu-Mi Lee, Ki-Ho Park, Mi Suk Lee.

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
