## [Decision Letter · Decision Letter 0]

23 Jun 2024

PONE-D-24-03240Visualization of multidrug-resistant bacterial infection trends in the intensive care unitsPLOS ONE

Dear Dr. Lee,

Thank you for submitting your manuscript to PLOS ONE. After careful consideration, we feel that it has merit but does not fully meet PLOS ONE’s publication criteria as it currently stands. Therefore, we invite you to submit a revised version of the manuscript that addresses the points raised during the review process.

We look forward to receiving your revised manuscript.

Kind regards,

Dwij Raj Bhatta, PhD

Academic Editor

PLOS ONE

Journal Requirements:

**Additional Editor Comments:**

Authors are sugested to adress coments of the reviewers.

Reviewers' comments:

Reviewer's Responses to Questions

**Comments to the Author**

1. Is the manuscript technically sound, and do the data support the conclusions?

Reviewer #1: No

Reviewer #2: Yes

2. Has the statistical analysis been performed appropriately and rigorously? 

Reviewer #1: I Don't Know

Reviewer #2: I Don't Know

3. Have the authors made all data underlying the findings in their manuscript fully available?

Reviewer #1: No

Reviewer #2: Yes

4. Is the manuscript presented in an intelligible fashion and written in standard English?

Reviewer #1: No

Reviewer #2: Yes

5. Review Comments to the Author

Reviewer #1: This is a descriptive study on multidrug-resistant bacterial infections in Korean Intensive Care Units through a new graphical transformation method.

MAJOR COMMENTS

•The main concern on the study, as also stated by the Authors, is the lack of a validation of the method for the graphical representation of the data. Authors defined this method as “new infection control indicator” but no evidence was provided.

•The Authors described the following graphical representation method: “An arrow was drawn with the average of the CS bacterial infection rate as the starting point and that of the CR bacterial infection rate as the endpoint. Then, the length and angle of the arrow were measured. In this process, the overall infection rate was scaled up 20 times”. However, not enough clear reasons for such a representation, and, particularly, mathematical basis for that were provided. How can readers understand the elements of representation and not see them as an arbitrary choice of the authors? What mathematical models underlie the graphic representation?

•Statistical method: Python script should be provided to better understand the study design and method for calculation.

•Criteria for stratification in red, yellow and green groups should be provided.

•Advantages of the graphical representation of the study compared to traditional methods should be largely discuss.

•Authors affirmed that such a graphical representation could be useful for “identifying factors for reducing infections” but this is not proven by the study.

MINOR COMMENTS

•The whole abstract is not enough clear

•Abstract, Results: reasons for the comparison between the first 5 years of observation and following 5-years are not understood (line 33)

•Introduction line 69: “Furthermore…”. This affirmation should be demonstrated.

•Line79: therefore, is it a prevalence study? And why the Authors have chosen to include only infections “that occurred on the third day after ICU admission”?

•CRBSI were not included into the study? Why not?

•Line 108: “ICUs where both CS and CR bacterial infections had occurred for more than two-quarters were finally selected.” Explain why.

• Line 102: The Authors stated: “Among the ICUs that participated in the KONIS from the 3rd quarter of 2006 to the 2nd quarter of 2022, those who participated in the first 5 years (3rd quarter of 2006 to 2nd quarter of 2011) and those who participated in the last 5 years (3rd quarter of 2017 to 2nd quarter of 2022) were selected”. Does it mean that two different groups of ICUs were selected? If so, Authors should clarify reasons and criteria for this selection method. Otherwise a clearer description of population stratification should be provided.

Reviewer #2: The manuscript is well written and findings are important in understanding the trends of nosocomial infections in ICU. The quality of the research paper can be improved by incorporating following points:

1. One of the most important bacterial nosocomial bacterial pathogen Staphylococcus aureus (MRSA) has not been included in this study. Details of MRSA may be included if data is available.

2. Duration of each quarter of the study is not clearly mentioned and difficult to understand.

3. Methods of detection of carbapenem resistance was not included in the methodology section.

4. Information provided in the figures is difficult to understand.

6. PLOS authors have the option to publish the peer review history of their article (what does this mean?). If published, this will include your full peer review and any attached files.

Reviewer #1: **Yes: **Rita Murri

Reviewer #2: No

---

## [Author Response · Author response to Decision Letter 1]

25 Aug 2024

please see the attached file named "Response to Reviewers.docx"

---

## [Decision Letter · Decision Letter 1]

17 Dec 2024

PONE-D-24-03240R1Visualization of multidrug-resistant bacterial infection trends in the intensive care unitsPLOS ONE

Dear Dr. Lee,

Thank you for submitting your manuscript to PLOS ONE. After careful consideration, we feel that it has merit but does not fully meet PLOS ONE’s publication criteria as it currently stands. Therefore, we invite you to submit a revised version of the manuscript that addresses the points raised during the review process. Please submit your revised manuscript by Jan 31 2025 11:59PM. If you will need more time than this to complete your revisions, please reply to this message or contact the journal office at plosone@plos.org. Please include the following items when submitting your revised manuscript:

We look forward to receiving your revised manuscript.

Kind regards,

Ali Amanati

Academic Editor

PLOS ONE

**Additional Editor Comments:**

Editor’s comments

Your manuscript [PONE-D-24-03240R1] has passed the review stage and is ready for ‎revision. ‎

To ensure the Editor and Reviewers can recommend that your revised manuscript be ‎accepted, ‎‎‎please pay careful attention to each comment posted underneath ‎this email. This way we ‎can ‎‎avoid future clarifications and revisions, moving swiftly to ‎a decision.‎

Technical points:‎

‎1. Please provide a point-by-point response to the Editor and reviewer's comments.

‎2. Please highlight all the amends on your manuscript with a yellow color‎.

‎3. Improve the English language of the manuscript‎

The Editor’s main concerns:‎

‎Request for Simplified Explanations of Figures

The authors can enhance the clarity of their figures and ensure that readers can easily understand the basis of the visual representations in the article.

Clarify the Purpose of Each Figure: The authors should explicitly state what each figure represents. They should explain the significance of the data shown and how it relates to the overall findings of the study. This will help readers grasp the context of the figures better.

Detailed Description of Vectors: The authors must provide a step-by-step explanation of how the vectors are calculated. This should include:

The specific data points used for calculations.

What formulas have been applied to derive the length and angle of the vectors?

An example calculation should be added to illustrate the process clearly.

It is highly encouraged that the authors include numerical examples alongside the figures. For instance, how do these rates translate into vector lengths and angles?

A visual representation of the calculations, perhaps in a supplementary table or diagram may be helpful.

The authors should consider adding annotations directly to the figures. This could include:

-Labels for key data points.

Reviewers' comments:

Reviewer's Responses to Questions

**Comments to the Author**

1. If the authors have adequately addressed your comments raised in a previous round of review and you feel that this manuscript is now acceptable for publication, you may indicate that here to bypass the “Comments to the Author” section, enter your conflict of interest statement in the “Confidential to Editor” section, and submit your "Accept" recommendation.

Reviewer #2: (No Response)

Reviewer #3: (No Response)

Reviewer #4: (No Response)

Reviewer #5: (No Response)

Reviewer #6: (No Response)

Reviewer #7: All comments have been addressed

Reviewer #8: All comments have been addressed

Reviewer #9: (No Response)

2. Is the manuscript technically sound, and do the data support the conclusions?

Reviewer #2: Yes

Reviewer #3: No

Reviewer #4: Partly

Reviewer #5: Partly

Reviewer #6: Yes

Reviewer #7: Yes

Reviewer #8: Partly

Reviewer #9: Yes

3. Has the statistical analysis been performed appropriately and rigorously? 

Reviewer #2: I Don't Know

Reviewer #3: Yes

Reviewer #4: I Don't Know

Reviewer #5: No

Reviewer #6: N/A

Reviewer #7: Yes

Reviewer #8: No

Reviewer #9: Yes

4. Have the authors made all data underlying the findings in their manuscript fully available?

Reviewer #2: Yes

Reviewer #3: Yes

Reviewer #4: No

Reviewer #5: No

Reviewer #6: No

Reviewer #7: Yes

Reviewer #8: Yes

Reviewer #9: Yes

5. Is the manuscript presented in an intelligible fashion and written in standard English?

Reviewer #2: Yes

Reviewer #3: No

Reviewer #4: Yes

Reviewer #5: Yes

Reviewer #6: Yes

Reviewer #7: Yes

Reviewer #8: Yes

Reviewer #9: Yes

6. Review Comments to the Author

**Reviewer #2:** The manuscript is well written and results are interesting. The duration of study is quite long with large number of hospital Intensive care units were included. Nosocomial infections are most commonly reported in the ICUs and are major cause of morbidity and mortality due to increasing incidences of antibiotic resistance. This study has incorporated three important nosocomial bacterial pathogens however, one common organism methicillin resistant Staphylococcus aureus (MRSA) has not been included. Study findings will be valuable in managing the increasing trends of multi drug resistant pathogens in ICUs. Overall impression of the study findings and quality of manuscript seems good.

**Reviewer #3:** The paper by Jeon et al. tries to compose an overview of infection rates and carbapenem-resistance in selected bacterial species in ICUs in Korea. This is no easy task and their efforts are worthwile, but the intention with the task is not completely clear. The data presented in the two tables are in my opinion the most interesting - both nationally and individually for the ICUs - since they permit a kind of trouble-shooting regarding a cause-relationship between factors of known importance for infection rates such as number of ICU-nurses/bed, number of infection control nurses/1000 beds, catheter days etc. Antibiotic consumption data would be very valuable in this context also.

Comments:

1. It is not clear to the reader whether the overview of infection rates and MDR-rates is meant for a central monitoring instituion (here KONIS) or also for the individual ICUs? Will each ICU receive a notofication of their standing in the monitoring proces, and is this annually, or do they only have access to the country-wide overview? The most logical would be the former, i.e. each ICU is notified withindividual and nationl data, which would benefit benchmarking?

2. The figure (fig. 4)with the circles and arrows marked as red spots (they must start in the central point in the circle?) is the most difficult to read and understand, while the figures with coloured areas in circle (green, yellow and red) are clearly most informative. Also difficult to read is Figure 1. It is a little confusing that short arrows are worst (rapid change), although the authors try to argue for this version.

3. BSI definition is problematuc: Bacteria in blood cannot be present in other infections? What about a K.penumoniae in urine or sputum, and in blood, which is usually the case; will it not be counted as a BSI?

4. Table 1: Size of ICU beds should be number of ICU beds. Infection nurse is usually called an infection control nurse.

5. Table 2: What do infection rates mean? Per number of patients or beddays or...? How is catheter-utilization or ventilator urilization ratio calculated?

6. Not all ICUs apparently have quarteers with iinfections with the 3 different bacterial species; this must also be an interesting number to follow? And why do they not have these infections - because they do not monitor or because they are better to treat their patients?

**Reviewer #4: **This is a novel method of representing the changes in multidrug-resistant microorganisms AB, KP, and PA by demonstrating the changes by arrows the graph transformation to demonstrate the multidrug-resistant bacterial infection trends in ICUs.

My queries on the paper are

1. Why was only one organism chosen per ICU, when multiple drug-resistant bacteria cause infection in ICUs?

2. How did the authors find arrow categories as useful tools in infection control, as compared to the conventional method, as there is no comparative data?

3. How did they identify factors for reducing infections?

4. As the authors have mentioned, there are several limitations to the study; it would be prudent to carry out a pilot study to compare changes in drug-resistant bacteria in ICUs by the conventional method to substantiate their claim.

**Reviewer #5: **Dear authors

Thank you for efforts put in addressing the concerns of the reviewers. Kindly have some consideration for my concerns below

In my opinion, the discussion could still be better. At least comparison of the current observatory methods proposed by the authors could be made with previous methods. No citation was made in the entire discussion.

Also, the authors could help readers by briefly summarizing at what points Student’s t-test, Mann

Whitney test, chi-square test, and Fisher’s exact test were employed. This concern is being raised because looking at the current results, it is not easy to see for instance at what point the skewed data analysis tool Mann Whitney test was employed and then the normality tools were also employed on the same data set.

**Reviewer #6:** I have not previously participated in rounds of review for this manuscript, but it seems to me that the authors have adressed previous comments. I find the paper interesting and relevant for epidemiologists and others working with surveillance of infectious diseases. The methods are sufficiently described to give an impression of how this tool may be used in different settings.

**Reviewer #7:** Comment to Authors

The graph transformation technique and the categorisation of ICUs into red, yellow, and green are novel and valuable contributions to the field. This study addresses a critical need in ICU infection control by simplifying extensive longitudinal surveillance data into actionable visual trends. The manuscript presents a thorough analysis of various pathogens and types of infections, utilising robust statistical methods and a careful selection of variables. The authors have effectively addressed the reviewers' concerns regarding their methods, enhancing the credibility of their findings. Additionally, they clarified significant aspects of their methodology based on the reviewers' feedback, such as their choice of scaling factors, the criteria for selecting which ICUs to include, and the thresholds used for categorization. Overall, the manuscript demonstrates that the authors have taken the reviewers' feedback seriously by making unclear sections more comprehensible and ensuring transparency in their methods.

Areas where additional clarity or improvements are still necessary:

1. Validation of Graph Transformation Technique:

The authors addressed concerns about the novelty of their visualization method, but the manuscript would benefit from a comparison to traditional methods. For example:

• How do the results from graph transformation compare to standard trend analysis methods, such as regression or time-series analysis?

• Do the ICU categories (red, yellow, green) yield the same conclusions in different validation datasets?

2. Interpretation of Polarization in Pseudomonas aeruginosa Trends:

While the authors note an unexpected rise in both "red" and "green" ICUs for P. aeruginosa, they do not explore potential reasons in depth. Providing specific ideas or analysing the characteristics of the ICUs could enhance this discussion.

3. Generality and Broader Implications:

The revised manuscript has improved its technical analysis but needs a more robust discussion about how the method can be applied across different healthcare systems. How well can this method adapt to systems with varying surveillance setups or different types of pathogens?

4. Addressing Reviewer Concerns About Bias:

Reviewers expressed concerns about potential bias due to excluded ICUs and the small sample size in earlier years. For instance, only three ICUs were analysed for Klebsiella pneumoniae in the first five years. The authors mention that the data is consistent but should discuss how these exclusions could impact the results. They should also clarify whether they conducted any sensitivity analyses.

5. Ethical Considerations and Data Accessibility:

Although the authors explained ethical approvals and how they maintained information privacy, replicating the study remains challenging due to limited data access. Including a synthetic dataset or providing clear instructions for repeating the analysis using publicly available data would enhance the manuscript's utility.

6. Writing and Presentation:

The revised manuscript is clearer, but some sections, particularly the introduction, remain overly wordy. Condensing these parts would improve readability.

Recommendations

a) If feasible, compare the graph transformation method with traditional methods. Including even a brief mention of ongoing validation efforts in the discussion would increase confidence in the approach i.e. the validity of the study.

b) Add a section discussing how this method could be adapted to other situations or pathogens, this will strengthen the generalizability of the study.

c) If feasible, provide a synthetic dataset or an example analysis pipeline in the supplementary materials to enable readers to apply the method to their own data.

d) Expand the discussion on the polarization phenomenon observed with Pseudomonas aeruginosa.

Conclusion

The manuscript demonstrates significant improvement and addresses most of the reviewers' concerns. Its new visualization method and insights into ICU infection trends make it a valuable addition to the field and is a strong candidate for publication. However, further discussion on validation, generalizability, and unexpected results would enhance its impact, if addressed.

**Reviewer #8:** (General comment) Please avoid the use of the same word more than once in a sentence. (e.g. "Graph transformation was employed to convert the infection rate graphs of each ICU into arrows." graph and rate graphs ...).

(General comment) Do not start a sentence with an abbreviation (e.g. UTI, BSI).

(General comment) Do not use abbreviations in section and subsections name (e.g., ICUs).

(General comment) To be clinical relevant you should also include information about the total number of patients hospitalized in ICU in the investigated period.

(General comment) It has sense to report median value for qualitative variable? For example, how do you interpret Pneumonia rate of 1.39? From my point of view is useless. The same comment is also valid for Q1 and Q3.

(General comment) One information should stay in only one place in the manuscript otherwise is duplicated (within manuscript duplication/triplication etc.). See for example "Significance and implications" section.

(Introduction) Avoid expressions such as "In recent years" in scientific writing.

(Introduction) Be specific when you say "increased" and let the reader from which number and when to which number - when.

(Methods) "The diseases analyzed were pneumonia, urinary tract infection (UTI), and bloodstream infection (BSI)." why these diseases?

(Methods) How many ICUs participated in KONIS?

(Methods) Briefly present the KONIS.

(Methods) "KONIS has collected and analyzed data on a quarterly basis according to its own guidelines" reference(s) is/are needed.

(Statistical analysis) Please tell the readers what method was used when.

(Results) "Among the selected ICUs, the causative pathogen was classified into CS and CR bacteria according to the carbapenem susceptibility status. ICUs where CS and CR bacterial infections occurred for more than two quarters were finally selected." & "For each selected ICU, an overall infection rate versus time graph was generated using Python. Subsequently, graph transformation was performed to create arrows for each ICU, with the endpoints indicated in polar coordinates." & "Each point on the graph signifies the endpoint of the arrow of an individual ICU, and the arrow for each ICU is formed by connecting the indicated point and origin. The arrows representing individual ICUs had various lengths and angles, with the diversity in .. "This information belongs to the methods section.

(Results) "However, since only three ICUs were analyzed in the first 5 years, caution should be observed when interpreting the results." This is not a result.

(Results) Pay attention to duplication of results in text and table/figure. A result must stay either in text or in table/figure.

(Results) If the value of Q1 and q3 are included in the ranges you should use squared brackets ("").

(Discussion) The shifts in trends must be explains (ex. more medical and less surgical in the second period compared to the first period ... why?).

(Discussion) The method you propose is at the seems to be useful at the level of hospital and you should discuss how these results could be used.

**Reviewer #9: **Title: - Visualization of multidrug-resistant bacterial infection trends in the intensive care unit.

I thank the authors for their innovative approach to visualizing infection trends and multidrug-resistant (MDR) gram-negative bacterial infections across numerous intensive care units (ICUs). This study addresses the complex challenge of simultaneously acquiring and interpreting surveillance data from hundreds of ICUs, which is crucial for enhancing infection control measures.

The well-structured abstract provides a transparent background, methods, results, and conclusions. The explanation of the graph transformation method is concise, but it might be too technical for readers unfamiliar with the concept. A brief explanation of why this method was chosen could enhance understanding, and it is better to emphasize the potential impact of the findings on healthcare practices and policies.

Introduction

The introduction should clearly articulate the research problem and the significance of the study. This is essential to ensure that the reader understands the research's context and necessity. A lack of clarity in the introduction may confuse readers about the study's purpose and scope.

The introduction should include a brief review of existing literature related to the study. This helps identify gaps in current knowledge and justify the need for the research. It is important to ensure that the literature review is up-to-date and relevant.

Method

The method section should provide sufficient detail to allow other researchers to replicate the study. This includes a comprehensive description of the data sources, the process of graph transformation, and how the infection rates were converted into arrows. The study uses data from the Korean National Healthcare-associated Infections Surveillance System (KONIS), covering 137 ICUs from 2006 to 2011 and 368 ICUs from 2017 to 2022. It is important to ensure that the selection criteria for these data sources are clearly explained, including any inclusion or exclusion criteria.

Discussion

While the discussion acknowledges the limitations, it could benefit from a deeper analysis of their potential implications. For instance, discussing how these limitations might affect the results' interpretation or the findings' generalizability would provide a more comprehensive understanding. The discussion could be strengthened by comparing the new method with existing methods for monitoring infection trends.

7. PLOS authors have the option to publish the peer review history of their article (what does this mean?). If published, this will include your full peer review and any attached files.

Reviewer #2: No

Reviewer #3: No

Reviewer #4: No

Reviewer #5: No

Reviewer #6: No

Reviewer #7: No

Reviewer #8: No

Reviewer #9: **Yes: **Helina Kurbi

---

## [Author Response · Author response to Decision Letter 2]

22 May 2025

The responses to reviewers and editor comments have been attached in the file named 'Response to Reviewers.docx.'

---

## [Decision Letter · Decision Letter 2]

22 Jun 2025

PONE-D-24-03240R2Visualization of multidrug-resistant bacterial infection trends in the intensive care unitsPLOS ONE

Dear Dr. Lee,

Thank you for submitting your manuscript to PLOS ONE. After careful consideration, we feel that it has merit but does not fully meet PLOS ONE’s publication criteria as it currently stands. Therefore, we invite you to submit a revised version of the manuscript that addresses the points raised during the review process. Please submit your revised manuscript by Aug 06 2025 11:59PM. If you will need more time than this to complete your revisions, please reply to this message or contact the journal office at plosone@plos.org. Please include the following items when submitting your revised manuscript:

We look forward to receiving your revised manuscript.

Kind regards,

*
**Ali Amanati**
*

**Academic Editor**

PLOS ONE

Journal Requirements:

Additional Editor Comments:

Dear authors, ‎

‎The invited reviewers (Reviewers #5 & #8) posted new comments. So, the manuscripts ‎‎require a ‎round of revision.‎ Please provide a point-by-point response to the ‎‎reviewers' ‎comments and highlight all the ‎amends on your manuscript with ‎‎yellow color.‎ ‎

Yours,

Reviewers' comments:

Reviewer's Responses to Questions

**Comments to the Author**

1. If the authors have adequately addressed your comments raised in a previous round of review and you feel that this manuscript is now acceptable for publication, you may indicate that here to bypass the “Comments to the Author” section, enter your conflict of interest statement in the “Confidential to Editor” section, and submit your "Accept" recommendation.

Reviewer #3: All comments have been addressed

Reviewer #4: All comments have been addressed

Reviewer #5: (No Response)

Reviewer #8: (No Response)

2. Is the manuscript technically sound, and do the data support the conclusions?

Reviewer #3: Yes

Reviewer #4: Yes

Reviewer #5: Partly

Reviewer #8: Partly

3. Has the statistical analysis been performed appropriately and rigorously? 

Reviewer #3: Yes

Reviewer #4: Yes

Reviewer #5: Yes

Reviewer #8: Yes

4. Have the authors made all data underlying the findings in their manuscript fully available?

Reviewer #3: No

Reviewer #4: Yes

Reviewer #5: No

Reviewer #8: No

5. Is the manuscript presented in an intelligible fashion and written in standard English?

Reviewer #3: Yes

Reviewer #4: Yes

Reviewer #5: Yes

Reviewer #8: Yes

6. Review Comments to the Author

Reviewer #3: The authors have responded adequately to my comments,. Figure 3 was not readable, just black square.

Reviewer #4: The authors have addressed all the queries and concerns by the reviewers satisfactorily.

They have made the necessary changes in the revised manuscript. Provided clarity and explanations to the figures and tables satisfactorily so that they are understood by the reader.

The representation of data by applying new statistical calculations and diagrammatic representation has been explained clearly.

Reviewer #5: Dear Authors

Thank you for the addressing some of the concerns raised and the attempt to address others.

I do not have further comments at this point

Reviewer #8: Abstract

- It is not sufficiently clear why you choose the two periods or bacteria.

- Is increase/decrease statistically significant?

Introduction

- Define abbreviations first time when are used in the body of the manuscript (CRAB. ICU, CRE).

- It is unclear why the reported prevalence is for US but the study is done on Korean data.

- How data are reported to KONIS?

- Is access to KONIS data free? How raw data can be accessed?

- "several studies using this information have been conducted [8]." if several it is expected to have more references.

- "have increased significantly" from which value to which values? in which period of time?

- Briefly introduce the process for ICUs monitored in KONIS. Are all ICUs in the country?

- It is not sufficiently clear if such evaluations were already conducted or not.

Material and methods

- Is KONIS data available from free to anyone? Please provide the source of data and describe how did you accessed it.

- How the disease were selected?

- "is generally low" please be specific.

- Why 2022? (we are in 2025).

Results

- "Among these, ICUs in which infections (pneumonia, UTI, and BSI) caused by KP, PA, and AB occurred in at least one quarter were selected." This sentence belongs to methods.

- Do not start a sentence with an abbreviation.

- Is the increased/decreased statistically significant?

- The graphical representations are blurred.

Discussion

- Please also discuss the generalizability of the reported results as well as the utility for ICUs and/or country.

Conclusions

- "Existing data was transformed through a series of processes to visualize it in a single graph, making it easier to understand the situation across hundreds of ICUs." This is not a conclusion.

"Data cannot be shared publicly because of potentially sensitive information" This declaration is opposite with "All participated intensive care unit’s data were fully anonymized before we accessed them".

7. PLOS authors have the option to publish the peer review history of their article (what does this mean?). If published, this will include your full peer review and any attached files.

Reviewer #3: No

Reviewer #4: No

Reviewer #5: No

Reviewer #8: No

---

## [Author Response · Author response to Decision Letter 3]

3 Jul 2025

We have revised our paper as described above and we believe we have addressed all questions and comment, but would be happy to provide further information or revision if necessary. Thank you for your consideration.

---

## [Editor Report · Decision Letter 3]

22 Jul 2025

PONE-D-24-03240R3Visualization of multidrug-resistant bacterial infection trends in the intensive care unitsPLOS ONE

Dear Dr. Lee,

Thank you for submitting your manuscript to PLOS ONE. After careful consideration, we feel that it has merit but does not fully meet PLOS ONE’s publication criteria as it currently stands. Therefore, we invite you to submit a revised version of the manuscript that addresses the points raised during the review process. Please submit your revised manuscript by Sep 05 2025 11:59PM. If you will need more time than this to complete your revisions, please reply to this message or contact the journal office at plosone@plos.org. Please include the following items when submitting your revised manuscript:

We look forward to receiving your revised manuscript.

Kind regards,

Ali Amanati

Academic Editor

PLOS ONE

Journal Requirements:

**Additional Editor Comments:**

Dear Authors,

Thank you for your revisions and your continued efforts to address the reviewer’s comments.

Because the reviewer has repeatedly raised questions regarding the study's setup and methodology, we kindly ask that you consolidate your responses to the following reviewer’s comments into a single, clearly organized section in your revised manuscript. Please assign an appropriate heading (e.g., "Setting and Study Design") to this section.

Specifically, please address and reorder your responses as follows:

Q1. (Abstract) It is not sufficiently clear why you chose the two periods or bacteria.

Q4. (Introduction) It is unclear why the reported prevalence is for US but the study is done on Korean data.

Q5. (Introduction) How are data reported to KONIS?

Q6. (Introduction) Is access to KONIS data free? How can raw data be accessed?

Q9. (Introduction) Briefly introduce the process for ICUs monitored in KONIS. Are all ICUs in the country?

Q11. (Material and methods) Is KONIS data available for free to anyone? Please provide the source of data and describe how you accessed it.

Q12. (Material and methods) How were the diseases selected?

Q14. (Material and methods) Why 2022? (We are in 2025).

Please make sure your responses are clear, concise, and complete for each point. This will help ensure clarity for the reviewer and streamline the review process.

Thank you very much for your cooperation.

Best regards,

---

## [Author Response · Author response to Decision Letter 4]

31 Jul 2025

We have revised our paper as described above and we believe we have addressed all questions and comment, but would be happy to provide further information or revision if necessary. Thank you for your consideration.

---

## [Editor Report · Decision Letter 4]

6 Aug 2025

Visualization of multidrug-resistant bacterial infection trends in the intensive care units

PONE-D-24-03240R4

Dear Dr. Mi Suk Lee,

We’re pleased to inform you that your manuscript has been judged scientifically suitable for publication and will be formally accepted for publication once it meets all outstanding technical requirements. Within one week, you’ll receive an e-mail detailing the required amendments. When these have been addressed, you’ll receive a formal acceptance letter and your manuscript will be scheduled for publication.

Kind regards,

Ali Amanati

Academic Editor

PLOS ONE

Additional Editor Comments (optional):

The authors have effectively utilized all available resources and data to enhance the manuscript, making it ‎more scientifically robust than before. Therefore, based on my opinion and the esteemed ‎reviewers' ‎‎comments, it could be published in its current form.‎

Yours‎,

---

## [Editor Report · Acceptance letter]

PONE-D-24-03240R4

PLOS ONE

Dear Dr. Lee,

I'm pleased to inform you that your manuscript has been deemed suitable for publication in PLOS ONE. Congratulations! Your manuscript is now being handed over to our production team.

Kind regards,

on behalf of

Professor Ali Amanati

Academic Editor

PLOS ONE